# GLOBAL MOMENTUM COMPRESSION
# FOR SPARSE COMMUNICATION IN DISTRIBUTED SGD

## ABSTRACT

With the rapid growth of data, distributed stochastic gradient descent (DSGD) has been widely used for solving large-scale machine learning problems. Due to the latency and limited bandwidth of network, communication has become the bottleneck of DSGD when we need to train large scale models, like deep neural networks. Communication compression with sparsified gradient, abbreviated as *sparse communication*, has been widely used for reducing communication cost in DSGD. Recently, there has appeared one method, called deep gradient compression (DGC), to combine memory gradient and momentum SGD for sparse communication. DGC has achieved promising performance in practice. However, the theory about the convergence of DGC is lack. In this paper, we propose a novel method, called *global momentum compression* (GMC), for sparse communication in DSGD. GMC also combines memory gradient and momentum SGD. But different from DGC which adopts local momentum, GMC adopts global momentum. We theoretically prove the convergence rate of GMC for both convex and non-convex problems. To the best of our knowledge, this is the first work that proves the convergence of distributed momentum SGD (DMSGD) with sparse communication and memory gradient. Empirical results show that, compared with the DMSGD counterpart without sparse communication, GMC can reduce the communication cost by approximately 100 fold with no loss of generalization accuracy. GMC can also achieve comparable (sometimes better) performance compared with DGC, with an extra theoretical guarantee.

## 1 INTRODUCTION

Many machine learning models can be formulated as the following empirical risk minimization problem:

$$\min_{\mathbf{w} \in \mathbb{R}^d} F(\mathbf{w}) := \frac{1}{n} \sum_{i=1}^{n} f(\mathbf{w}; \xi_i), \tag{1}$$

where $\mathbf{w}$ denotes the model parameter, $\xi_i$ denotes the $i$th training data, $n$ is the number of training data, and $d$ is the size of the model. For example, let $\xi_i = (\mathbf{a}_i, y_i)$, where $\mathbf{a}_i$ denotes the feature of the $i$th training data and $y_i$ denotes the label. Then in logistic regression $f(\mathbf{w}; \xi_i) = \log(1 + e^{-y_i \mathbf{a}_i^T \mathbf{w}}) + \frac{\lambda}{2}\|\mathbf{w}\|^2$, and in SVM $f(\mathbf{w}; \xi_i) = \max(0, 1 - y_i \mathbf{a}_i^T \mathbf{w}) + \frac{\lambda}{2}\|\mathbf{w}\|^2$. Many deep learning models can also be formulated as (1).

One of the efficient ways to solve (1) is stochastic gradient descent (SGD) (Robbins & Monro, 1951). In each iteration, SGD calculates one stochastic gradient $\nabla f(\mathbf{w}; \xi_i)$ and updates $\mathbf{w}$ by $\mathbf{w} \leftarrow \mathbf{w} - \eta \nabla f(\mathbf{w}; \xi_i)$, or updates $\mathbf{w}$ with a mini-batch of stochastic gradients. Inspired by momentum and nesterov's accelerated gradient descent, momentum SGD (MSGD) (Polyak, 1964; Tseng, 1998; Lan, 2012; Kingma & Ba, 2015) has been proposed and widely used in machine learning. In practice, MSGD can achieve better performance than SGD (Krizhevsky et al., 2012; Sutskever et al., 2013). Many machine learning platforms like TensorFlow, PyTorch, and MXNet adopt MSGD as one of the optimization methods.

With the rapid growth of data, distributed SGD (DSGD) (Dekel et al., 2012; Li et al., 2014b) has attracted much attention since it can parallelly calculate a batch of stochastic gradients. DSGD can

be formulated as follows:

$$\mathbf{w}_{t+1} = \mathbf{w}_t - \frac{\eta_t}{p} \sum_{k=1}^{p} \mathbf{g}_{t,k}, \tag{2}$$

where $p$ is the number of workers, $\mathbf{g}_{t,k}$ is the stochastic gradient or a mini-batch of stochastic gradients calculated by the $k$th worker at the $t$th iteration. DSGD can be implemented on distributed frameworks like parameter server and all-reduce framework. Each worker calculates $\mathbf{g}_{t,k}$ and sends it to the server or other workers for updating $\mathbf{w}$. Recently, more and more large models, such as deep learning models, are used in machine learning to improve the generalization ability. In large models, $\mathbf{g}_{t,k}$ is a high dimensional vector. Due to the latency and limited bandwidth of network, communication cost has become the bottleneck of traditional DSGD or distributed MSGD (DMSGD). For example, when we implement DSGD on parameter server, the server needs to receive $p$ high dimension vectors from workers, which will lead to communication traffic jam and make the convergence of DSGD slow. Hence, we need to compress $\mathbf{g}_{t,k}$ to reduce the communication cost.

Recently, researchers have proposed two main categories of communication compression techniques for reducing communication cost in DSGD and DMSGD. The first category is quantization (Wen et al., 2017; Alistarh et al., 2017; Jiang & Agrawal, 2018). In machine learning problems, 32-bit float number is typically adopted for representation. Quantization methods quantize the value (gradient or parameter) representation from 32 bits to some low bit-width like 8 bits or 4 bits. Since the quantized gradients in most methods are an unbiased estimation of the original ones, the convergence rate of these methods has the same order of magnitude as that of DSGD, but slower due to the extra quantization variance. It is easy to find that the communication cost can be reduced by 31 fold in the ideal case. In practice, at least 4 bits should be adopted for representation in most cases to keep original accuracy. In these cases, the communication cost is reduced by 7 fold.

The other category is based on sparsified gradient, which is called *sparse communication*. In sparse communication, after calculating the update vector $\mathbf{g}_{t,k}$ at each iteration, each worker only sends a subset of coordinates in $\mathbf{g}_{t,k}$, denoted as $S(\mathbf{g}_{t,k})$, to the server or other workers. Here, $S(\mathbf{g}_{t,k})$ is a sparse vector and hence it can reduce the communication cost. In recent works (Aji & Heafield, 2017; Lin et al., 2018), each worker will typically remember those values which are not sent to the server, i.e., $\mathbf{g}_{t,k} - S(\mathbf{g}_{t,k})$, and store it in the *memory* rather than dropping them. The $\mathbf{g}_{t,k} - S(\mathbf{g}_{t,k})$ is called *memory gradient* and will be used to calculate the next update vector $\mathbf{g}_{t+1,k}$. This is intuitively necessary because a subset of coordinates in the stochastic gradient can not reflect the real descent direction and can make errors with higher probability than the original stochastic gradient. This memory gradient based sparse communication strategy has been widely adopted by recent communication compression methods and has achieved better performance than quantization methods and other sparse communication methods without memory gradient. In these memory gradient based sparse communication methods, some are for vanilla SGD (Aji & Heafield, 2017; Alistarh et al., 2018; Stich et al., 2018; Karimireddy et al., 2019; Tang et al., 2019). The convergence rate of vanilla SGD with sparse communication has been proved in (Alistarh et al., 2018; Stich et al., 2018; Karimireddy et al., 2019; Tang et al., 2019). Very recently, there has appeared one sparse communication method for distributed MSGD (DMSGD), called deep gradient compression (DGC) (Lin et al., 2018), which has achieved better performance than vanilla DSGD with sparse communication in practice. However, the theory about the convergence of DGC is still lack. Furthermore, although DGC uses momentum SGD, the momentum in DGC is calculated by each worker. Hence it is a local momentum without global information.

In this paper, we propose a novel method, called _g_lobal _m_omentum _c_ompression (GMC), for sparse communication in DMSGD which includes DSGD as a special case. The main contributions of this paper are summarized as follows:

- GMC combines memory gradient and momentum SGD to achieve sparse communication in DMSGD (DSGD). But different from DGC which adopts local momentum, GMC adopts global momentum.

- We theoretically prove the convergence rate of GMC for both convex and non-convex problems. To the best of our knowledge, this is the first work that proves the convergence of DMSGD with sparse communication and memory gradient.

- Empirical results show that, compared with the DMSGD counterpart without sparse communication, GMC can reduce the communication cost by approximately 100 fold with no loss of generalization accuracy.
- GMC can also achieve comparable (sometimes better) performance with comparable communication compression ratio, with an extra theoretical guarantee.

## 2 PRELIMINARY

In this paper, we use $\|\cdot\|$ to denote $L_2$ norm, use $\mathbf{w}^*$ to denote the optimal solution of (1), use $\nabla f(\mathbf{w}; \mathcal{I}) \triangleq \frac{1}{|\mathcal{I}|} \sum_{\xi_i \in \mathcal{I}} \nabla f(\mathbf{w}; \xi_i)$ to denote one stochastic gradient with respect to a mini-batch of samples $\mathcal{I}$ such that $\mathbb{E}_{\mathcal{I}}[\nabla f(\mathbf{w}; \mathcal{I})|\mathbf{w}] = \nabla F(\mathbf{w})$, use $\odot$ to denote element-wise product, use $\mathbf{1}$ to denote the vector $(1, 1, \ldots, 1)^T \in \mathbb{R}^d$, use $\mathbf{e}_j$ to denote the vector $(0, \ldots, 1, \ldots 0)^T \in \mathbb{R}^d$ with only the $j$th coordinate being 1, use $\mathbf{I}$ to denote an identity matrix. For a vector $\mathbf{a}$, we use $a^{(j)}$ to denote its $j$th coordinate value. $\|\mathbf{a}\|_0$ denotes the number of non-zero values in $\mathbf{a}$.

**Definition 1** *(smooth function) Function $h(\cdot)$ is L-smooth ($L > 0$) if $\forall \mathbf{w}, \mathbf{w}'$,*

$$|h(\mathbf{w}) - h(\mathbf{w}') - \nabla h(\mathbf{w}')^T(\mathbf{w} - \mathbf{w}')| \leq \frac{L}{2}\|\mathbf{w} - \mathbf{w}'\|^2.$$

**Definition 2** *(convex function) Function $h(\cdot)$ is convex if $\forall \mathbf{w}, \mathbf{w}'$,*

$$|h(\mathbf{w}) - h(\mathbf{w}') - \nabla h(\mathbf{w}')^T(\mathbf{w} - \mathbf{w}')| \geq \frac{\mu}{2}\|\mathbf{w} - \mathbf{w}'\|^2,$$

*where $\mu \geq 0$. If $\mu > 0$, $h(\cdot)$ is called a $\mu$-strongly convex function.*

### 2.1 DISTRIBUTED MOMENTUM SGD

The widely used momentum SGD (MSGD) (Polyak, 1964) for solving (1) can be written as

$$\mathbf{g}_t = \beta \mathbf{g}_{t-1} + \eta_t \nabla f(\mathbf{w}_t; \mathcal{I}_t),$$
$$\mathbf{w}_{t+1} = \mathbf{w}_t - \mathbf{g}_t,$$

where $\beta \in [0, 1)$. The $\mathbf{g}_t$ is the Polyak's momentum and $\nabla f(\mathbf{w}_t; \mathcal{I}_t)$ is an unbiased estimation of $\nabla F(\mathbf{w}_t)$. Since $\mathbf{g}_t = (\mathbf{w}_t - \mathbf{w}_{t+1})$, MSGD can also be written as

$$\mathbf{w}_{t+1} = \mathbf{w}_t - \eta_t(\nabla f(\mathbf{w}_t; \mathcal{I}_t) + \frac{\beta}{\eta_t}(\mathbf{w}_{t-1} - \mathbf{w}_t)). \quad (3)$$

Please note that if $\beta = 0$, MSGD degenerates to SGD.

One simple way to implement distributed MSGD (DMSGD) is that each worker parallelly calculates some stochastic gradients and then the stochastic gradients of all workers are aggregated to get $\nabla f(\mathbf{w}_t; \mathcal{I}_t) + \beta(\mathbf{w}_{t-1} - \mathbf{w}_t)/\eta_t$. The update process of $\mathbf{w}$ in this way is equivalent to the serial MSGD. We call $(\mathbf{w}_{t-1} - \mathbf{w}_t)/\eta_t$ the *global momentum*, because it captures the global information from all workers.

Another way to implement DMSGD is using *local momentum*:

$$\mathbf{g}_{t,k} = \beta \mathbf{g}_{t-1,k} + \eta_t \mathbf{v}_{t,k}, k = 1, 2, \ldots, p,$$
$$\mathbf{w}_{t+1} = \mathbf{w}_t - \sum_{k=1}^{p} \mathbf{g}_{t,k},$$

where $\mathbf{v}_{t,k}$ is the stochastic gradient calculated by the $k$th worker and $\sum_{k=1}^{p} \mathbf{v}_{t,k} = \nabla f(\mathbf{w}_t; \mathcal{I}_t)$. $\mathbf{g}_{t-1,k}$ is the *local momentum*. Since $\sum_{k=1}^{p} \mathbf{g}_{t-1,k} = (\mathbf{w}_{t-1} - \mathbf{w}_t)$, this DMSGD with local momentum can also be written as the formulation in (3). Hence, the global momentum contains all information of the local momentums. We will find that DGC (Lin et al., 2018) is mainly based on the local momentum while GMC is mainly based on the global momentum. Hence, each worker in DGC cannot capture the global information from its own local momentum, while that in GMC can capture the global information from the global momentum even if sparse communication is adopted. In the later section, we will see that global momentum is better than local momentum when using memory gradient for sparse communication.

---

**Algorithm 1** Global Momentum Compression (GMC) on Parameter Server

---

1: Initialization: $p$ workers, $\mathbf{w}_{-1} = \mathbf{w}_0$, $\beta \in [0, 1)$, batch size $b$;
2: Set $\mathbf{g}_{0,k} = \mathbf{u}_{0,k} = 0, k = 1, \ldots, p$,
3: **for** $t = 0, 1, 2, \ldots T - 1$ **do**
4:     Workers:
5:     **for** $k = 1, 2 \ldots, p$, **each worker parallelly do**
6:       **if** $t > 0$ **then**
7:         Receive $\mathbf{w}_t - \mathbf{w}_{t-1}$ from the server;
8:         Get $\mathbf{w}_t$ by $\mathbf{w}_t = \mathbf{w}_{t-1} + (\mathbf{w}_t - \mathbf{w}_{t-1})$;
9:       **end if**
10:       Randomly pick a mini-batch of training data $\mathcal{I}_{t,k} \subseteq D_k$ with $|\mathcal{I}_{t,k}| = b$;
11:       $\mathbf{g}_{t,k} = \frac{1}{pb} \sum_{\xi_i \in \mathcal{I}_{t,k}} \nabla f(\mathbf{w}_t; \xi_i) - \frac{\beta}{p\eta_t}(\mathbf{w}_t - \mathbf{w}_{t-1})$;
12:       Generate a sparse vector $\mathbf{m}_{t,k} \in \{0, 1\}^d$;
13:       Send $\mathbf{m}_{t,k} \odot (\mathbf{g}_{t,k} + \mathbf{u}_{t,k})$ to the server;
14:       $\mathbf{u}_{t+1,k} = (\mathbf{1} - \mathbf{m}_{t,k}) \odot (\mathbf{g}_{t,k} + \mathbf{u}_{t,k})$;
15:     **end for**
16:     Server:
17:     $\mathbf{w}_{t+1} = \mathbf{w}_t - \eta_t \sum_{k=1}^{p} \mathbf{m}_{t,k} \odot (\mathbf{g}_{t,k} + \mathbf{u}_{t,k})$;
18:     Send $\mathbf{w}_{t+1} - \mathbf{w}_t$ to workers;
19: **end for**

---

## 3   Global Momentum Compression

Assume we have $p$ workers. $D_k$ denotes the data stored on the $k$th worker ($k = 1, 2, \ldots, p$) and $\bigcup_{k=1}^{p} D_k = \{\xi_1, \xi_2, \ldots, \xi_n\}$. Our method Global Momentum Compression (GMC) mainly performs the following operations iteratively:

- Each worker calculates $\mathbf{g}_{t,k} = \frac{1}{pb} \sum_{\xi_i \in \mathcal{I}_{t,k}} \nabla f(\mathbf{w}; \xi_i) - \frac{\beta}{p\eta_t}(\mathbf{w}_t - \mathbf{w}_{t-1})$, where $\mathcal{I}_{t,k} \subseteq D_k$ and $|\mathcal{I}_{t,k}| = b$;

- Each worker generates a sparse vector $\mathbf{m}_{t,k}$ and sends $\mathbf{m}_{t,k} \odot (\mathbf{g}_{t,k} + \mathbf{u}_{t,k})$ to the server or other workers;

- Each worker updates $\mathbf{u}_{t+1,k} = (\mathbf{1} - \mathbf{m}_{t,k}) \odot (\mathbf{g}_{t,k} + \mathbf{u}_{t,k})$;

- Update parameter $\mathbf{w}_{t+1} = \mathbf{w}_t - \eta_t \sum_{k=1}^{p} \mathbf{m}_{t,k} \odot (\mathbf{g}_{t,k} + \mathbf{u}_{t,k})$;

Below, we introduce the framework and two essential elements of GMC: memory gradient $\mathbf{u}_{t,k}$ and global momentum $(\mathbf{w}_{t-1} - \mathbf{w}_t)/\eta_t$.

### 3.1   Framework of GMC

GMC can be easily implemented on the all-reduce distributed framework in which each worker sends the sparse vector $\mathbf{m}_{t,k} \odot (\mathbf{g}_{t,k} + \mathbf{u}_{t,k})$ to all the other workers, then each worker updates $\mathbf{w}_{t+1}$ after receiving the sparse vectors from other workers.

Recently, parameter server (Li et al., 2014a) has been one of the most popular distributed frameworks in machine learning. GMC can also be implemented on parameter server. Although the theories in this paper can be adopted for the all-reduce framework, in this paper we adopt parameter server for illustration. The details of GMC implemented on parameter server are shown in Algorithm 1. The difference between GMC and traditional DSGD on parameter server is that in GMC after updating $\mathbf{w}_{t+1}$, server will send $\mathbf{w}_{t+1} - \mathbf{w}_t$, rather than $\mathbf{w}_{t+1}$, to workers. Since $\mathbf{m}_{t,k}$ is sparse, $\mathbf{w}_{t+1} - \mathbf{w}_t$ is sparse as well. Hence, sending $\mathbf{w}_{t+1} - \mathbf{w}_t$ can reduce the communication cost compared with sending $\mathbf{w}_t$. In our experiments, we find that GMC can make $\|\mathbf{w}_{t+1} - \mathbf{w}_t\|_0 \leq 0.01d$ with no loss of accuracy when training large scale models. Workers can get $\mathbf{w}_{t+1}$ by $\mathbf{w}_{t+1} = \mathbf{w}_t + (\mathbf{w}_{t+1} - \mathbf{w}_t)$.

**Remark 1** *We can also use the sparse communication technique on the server (denoted as GMC\*) to make $(\mathbf{w}_{t+1} - \mathbf{w}_t)$ sparser than that in GMC so that when the server sends $\mathbf{w}_{t+1} - \mathbf{w}_t$ to workers, the communication cost can be further reduced, compared with GMC. The convergence of GMC\**

*can also be theoretically proved. But in practice, GMC\* will be slightly worse than GMC in terms of accuracy. Due to the space limitation, we move both theory and experiments about GMC\* to Appendix A.*

## 3.2 NECESSITY OF MEMORY GRADIENT

In GMC, after sending a sparse vector $\mathbf{m}_{t,k} \odot (\mathbf{g}_{t,k} + \mathbf{u}_{t,k})$ to the server, each worker will remember the coordinates which are not sent and store them in $\mathbf{u}_{t+1,k}$:

$$\mathbf{u}_{t+1,k} = (\mathbf{1} - \mathbf{m}_{t,k}) \odot (\mathbf{g}_{t,k} + \mathbf{u}_{t,k}). \tag{4}$$

Hence we call $\mathbf{u}_{t,k}$ the *memory gradient*, which is important for the convergence guarantee of GMC. Here, we give an intuitive explanation about why GMC needs to remember the coordinate values which are not sent. We consider the simple case that $\beta = 0$, which means $\mathbf{g}_{t,k}$ is a stochastic gradient of $F(\mathbf{w})$ and GMC degenerates to (Aji & Heafield, 2017). Since $\mathbf{m}_{t,k}$ is a sparse vector, GMC can be seen as a method achieving sparse communication by combining stochastic coordinate descent (SCD) (Nesterov, 2012) and DSGD. In SCD, each $-\nabla F(\mathbf{w})^{(j)}\mathbf{e}_j$ denotes a true descent direction. When we use a stochastic gradient $\nabla f(\mathbf{w}; \mathcal{I})$ to replace $\nabla F(\mathbf{w})$, $-\nabla f(\mathbf{w}; \mathcal{I})^{(j)}\mathbf{e}_j$ will make errors with high probability, especially when $\mathbf{m}_{t,k}$ adopts the strategy that choosing $s$ coordinates with the largest absolute values (denoted as top-$s$ strategy).

For further explaining the importance of memory gradient, we consider the following simple example: let $n = p = 2$, $f(\mathbf{w}; \xi_1) = (-\alpha, \epsilon)\mathbf{w}$, $f(\mathbf{w}; \xi_2) = (\alpha + \epsilon, \gamma)\mathbf{w}$, where $\mathbf{w} \in [-1, 0] \times [-1, 0], 0 < \epsilon < \alpha < \gamma < \alpha + \epsilon$. $f(\mathbf{w}; \xi_1)$ is on the first worker and $f(\mathbf{w}; \xi_2)$ is on the second worker. Then we run GMC with $\beta = 0$ to solve $\min F(\mathbf{w}) = \frac{1}{2}(f(\mathbf{w}; \xi_1) + f(\mathbf{w}; \xi_2))$. The optimal solution is $\mathbf{w}^* = (-1, -1)^T$. We adopt the top-1 strategy for $\mathbf{m}_{t,k}$ in this example.

If we do not use the memory gradient, which means each worker directly sends $\mathbf{m}_{t,k} \odot \mathbf{g}_{t,k}$ to the server, then the two workers will send $(-\alpha/2, 0)^T$ and $((\alpha + \epsilon)/2, 0)^T$ respectively to the server. The parameter is updated by $\mathbf{w} \leftarrow \mathbf{w} - \eta_t(\epsilon/2, 0)$. We observe that $w^{(2)}$ will never be updated. This is due to the pseudo large gradient value which cheats $\mathbf{m}_{t,k}$. Since $\nabla F(\mathbf{w}) = (\epsilon/2, (\gamma + \epsilon)/2)^T$, we can see that the second coordinate has the true large gradient value and we should have mainly focused on updating $w^{(2)}$. However, in the two stochastic gradients $\nabla f(\mathbf{w}; \xi_1)$ and $\nabla f(\mathbf{w}; \xi_2)$, the first coordinate has larger absolute value. Hence, $\nabla f(\mathbf{w}; \xi_k)$ cheats $\mathbf{m}_{t,k}$ which leads to the error.

If we use memory gradient, at the beginning, $\mathbf{m}_{t,1} = \mathbf{m}_{t,2} = (1, 0)^T$. After some iterations, $\mathbf{m}_{t,1} = (0, 1)^T$ and $\mathbf{m}_{t,2} = (0, 1)^T$ due to the memory gradient. Specifically, let $t_1, t_2$ be two integers satisfying $\alpha/\epsilon \leq t_1 < \alpha/\epsilon + 1, (\alpha + \epsilon)/\gamma \leq t_2 < (\alpha + \epsilon)/\gamma + 1$, then it is easy to verify that $\mathbf{m}_{ct_1,1} = (0, 1)^T, \mathbf{m}_{ct_2,2} = (0, 1)^T, \forall c \geq 1$. It implies that if we use the memory gradient, both $w^{(1)}$ and $w^{(2)}$ will be updated. Hence, GMC can make $\mathbf{w}$ converge to the optimum $(-1, -1)^T$.

Hence, the memory gradient is necessary for sparse communication. It can overcome the disadvantage of combining DSGD and SCD.

## 3.3 BENEFIT OF GLOBAL MOMENTUM

In GMC, each worker calculates $\mathbf{g}_{t,k}$ as

$$\mathbf{g}_{t,k} = \frac{1}{pb} \sum_{\xi_i \in \mathcal{I}_{t,k}} \nabla f(\mathbf{w}_t; \xi_i) - \frac{\beta}{p\eta_t}(\mathbf{w}_t - \mathbf{w}_{t-1}). \tag{5}$$

When $\beta = 0$, it degenerates to DSGD with sparse communication, which is also called gradient dropping in (Aji & Heafield, 2017). We use $\mathbf{g}_{t,k}^{(\text{GD})}$ to denote this degenerated version with $\beta = 0$. We can see that GMC uses the global momentum $(\mathbf{w}_{t-1} - \mathbf{w}_t)/\eta_t$. While in DGC (Lin et al., 2018), the $\mathbf{g}_{t,k}^{(\text{DGC})}$ is calculated by

$$\mathbf{g}_{t,k}^{(\text{DGC})} = \frac{1}{pb} \sum_{\xi_i \in \mathcal{I}_{t,k}} \nabla f(\mathbf{w}_t; \xi_i) + \beta \mathbf{g}_{t-1,k}^{(\text{DGC})}, \tag{6}$$

which uses the local momentum $\mathbf{g}_{t-1,k}^{(\text{DGC})}$. We use $\mathbf{u}_{t,k}^{(\text{DGC})}$ to denote the memory gradient in DGC.

For sparse $\mathbf{m}_{t,k}$, we can find that $\mathbf{g}_{t,k}^{(\text{DGC})}$ and $\mathbf{u}_{t,k}^{(\text{DGC})}$ only contain the information based on the local data, while $\mathbf{g}_{t,k}$ and $\mathbf{u}_{t,k}$ can contain the global information due to the global momentum. Hence, global momentum is better than local momentum when using memory gradient for sparse communication. Assume that $\mathbb{E}[F(\mathbf{w}_t)]$ converges to $F(\mathbf{w}^*)$, then $\mathbf{w}_t - \mathbf{w}_{t-1}$ denotes the descent direction with high probability. Since $\mathcal{I}_{t,k} \subseteq D_k$, which only contains partial information of the whole training data, the global momentum $(\mathbf{w}_{t-1} - \mathbf{w}_t)/\eta_t$ can make compensation for the error between the stochastic gradient and the full gradient. Specifically, if $(\mathbf{w}_t - \mathbf{w}_{t-1})^T(-\nabla F(\mathbf{w}_t)) \geq 0$, then we get that

$$
\begin{aligned}
\mathbf{g}_{t,k}^T \nabla F(\mathbf{w}_t) =& (\frac{1}{pb} \sum_{\xi_i \in \mathcal{I}_{t,k}} \nabla f(\mathbf{w}_t; \xi_i) - \frac{\beta}{p\eta_t}(\mathbf{w}_t - \mathbf{w}_{t-1}))^T \nabla F(\mathbf{w}_t) \\
\geq& \frac{1}{pb} \sum_{\xi_i \in \mathcal{I}_{t,k}} \nabla f(\mathbf{w}_t; \xi_i)^T \nabla F(\mathbf{w}_t) \\
=& (\mathbf{g}_{t,k}^{(\text{GD})})^T \nabla F(\mathbf{w}_t).
\end{aligned}
$$

It implies that $\mathbf{g}_{t,k}$ is a better estimation of $\nabla F(\mathbf{w}_t)$ than $\mathbf{g}_{t,k}^{(\text{GD})}$. The experiments in (Lin et al., 2018) also show that (Aji & Heafield, 2017) has some loss of accuracy with $\|\mathbf{m}_{t,k}\|_0 \approx 0.001d$, while we can set such a sparsity for GMC with no loss of accuracy in practice. Hence, GMC is better than (Aji & Heafield, 2017).

**Remark 2** *There are some other ways to combine global momentum and memory gradient. For example, we can put the global momentum on the server. However, these ways lead to some loss of performance. More discussions are put in Appendix B.*

## 4   CONVERGENCE OF GMC

In this section, we prove the convergence rate of GMC for both convex and non-convex problems. The detailed proofs are put in Appendix C. We define a diagonal matrix $\mathbf{M}_{t,k} \in \mathbb{R}^{d \times d}$ such that $\text{diag}(\mathbf{M}_{t,k}) = \mathbf{m}_{t,k}$ to replace the symbol $\odot$. Then we obtain

$$
\mathbf{w}_{t+1} = \mathbf{w}_t - \eta_t \sum_{k=1}^p \mathbf{M}_{t,k}(\mathbf{g}_{t,k} + \mathbf{u}_{t,k}), \tag{7}
$$

$$
\mathbf{u}_{t+1,k} = (\mathbf{I} - \mathbf{M}_{t,k})(\mathbf{g}_{t,k} + \mathbf{u}_{t,k}), k = 1, 2, \ldots, p. \tag{8}
$$

For convenience, we denote $\nabla f(\mathbf{w}; \mathcal{I}_t) = \frac{1}{pb} \sum_{k=1}^p \sum_{\xi_i \in \mathcal{I}_{t,k}} \nabla f(\mathbf{w}_t; \xi_i)$, $\tilde{\mathbf{u}}_t = \sum_{k=1}^p \mathbf{u}_{t,k}$, $\tilde{\mathbf{g}}_t = \sum_{k=1}^p \mathbf{g}_{t,k}$. By eliminating $\mathbf{M}_{t,k}$, we obtain

$$
\mathbf{w}_{t+1} = \mathbf{w}_t - \eta_t \nabla f(\mathbf{w}_t; \mathcal{I}_t) + \beta(\mathbf{w}_t - \mathbf{w}_{t-1}) - \eta_t \tilde{\mathbf{u}}_t + \eta_t \tilde{\mathbf{u}}_{t+1}. \tag{9}
$$

We can find that if we do not use the sparse communication technique, then $\tilde{\mathbf{u}}_t = 0$ and (9) is the same as momentum SGD in (3).

First, we give the following lemma:

**Lemma 1** *Let $\mathbf{x}_t = \mathbf{w}_t + \frac{\beta}{1-\beta}(\mathbf{w}_t - \mathbf{w}_{t-1}) - \frac{\eta_t}{1-\beta}\tilde{\mathbf{u}}_t$. Then we have:*

$$
\mathbf{x}_{t+1} = \mathbf{x}_t - \frac{\eta_t}{1-\beta} \nabla f(\mathbf{w}_t; \mathcal{I}_t) + \frac{\eta_t - \eta_{t+1}}{1-\beta}\tilde{\mathbf{u}}_{t+1}. \tag{10}
$$

Equation (10) is similar to the update equation in SGD except the additional term $\tilde{\mathbf{u}}_{t+1}$. Inspired by this, we only need to prove the convergence of $\mathbf{x}_t$ and $\|\mathbf{x}_t - \mathbf{w}_t\|$. To get the convergence results, we further make the following three assumptions:

**Assumption 1** *(bounded gradient)* $\mathbb{E}\|\nabla f(\mathbf{w}_t; \xi_i)\|^2 \leq G^2, \forall t$.

**Assumption 2** *(bounded memory)* $\mathbb{E}\|\tilde{\mathbf{u}}_t\|^2 \leq U^2, \forall t$.

**Assumption 3** *(bounded parameter)* $\mathbb{E}\|\mathbf{x}_t - \mathbf{w}^*\|^2 \leq D^2, \forall t$.

**Remark 3** *Assumption 1 is common in stochastic optimization. Assumption 2 is easy to be guaranteed. We discuss it in Appendix D. Assumption 3 is only for convenience in the analysis for strongly convex and convex cases. We can add one projection operation on Equation (7) to guarantee Assumption 3 (if both $\mathbf{w}_t$ and $\tilde{\mathbf{u}}_t$ are bounded, $\mathbf{x}_t$ is bounded as well), which can be written as $\mathbf{w}_{t+1} = \Pi_\Omega(\mathbf{w}_t - \eta_t \sum_{k=1}^p \mathbf{M}_{t,k}(\mathbf{g}_{t,k} + \mathbf{u}_{t,k}))$. The convergence can still be guaranteed if we use the projection operation. More details are put in Appendix E.*

For the new variable $\mathbf{x}_t$ in Lemma 1, the gap between $\mathbf{x}_t$ and $\mathbf{w}_t$ has the following property:

**Lemma 2** *With Assumption 1 and Assumption 2, we get that*

- *If $\eta_t = \eta$, then*

$$\mathbb{E}\|\mathbf{x}_t - \mathbf{w}_t\|^2 \leq \left[\frac{2\beta^2(2G^2 + 2U^2)}{(1-\beta)^4} + \frac{2U^2}{(1-\beta)^2}\right]\eta^2. \tag{11}$$

- *If $\eta_t = \frac{r}{(t+q)^\rho}, \rho \in [0.5, 1], r > 0, q \geq 0$, then*

$$\mathbb{E}\|\mathbf{x}_t - \mathbf{w}_t\|^2 \leq \left[\frac{2\beta^2(2G^2 + 2U^2)\max\{(1-\beta)A, 2\}}{(1-\beta)^4} + \frac{2U^2}{(1-\beta)^2}\right]\eta_t^2. \tag{12}$$

*where $A = \max_{t=0}^{t_0}\{\frac{\eta_t^2 + \beta\eta_{t-1}^2 + \ldots + \beta^t\eta_0^2}{\eta_{t+1}^2}\}$, $t_0$ satisfies $(\frac{t_0+1+q}{t_0+2+q})^{2\rho} \geq \frac{1+\beta}{2}$.*

Lemma 2 implies that $\mathbb{E}\|\mathbf{x}_t - \mathbf{w}_t\|^2 \leq \mathcal{O}(\eta_t^2)$, i.e. the distance between $\mathbf{x}_t$ and $\mathbf{w}_t$ is $\mathcal{O}(\eta_t)$. For convenience, below we use the constant $C_{r,q,\rho,\beta,\eta}$ to denote the upper bound of $\mathbb{E}\|\mathbf{x}_t - \mathbf{w}_t\|^2/\eta_t^2$. If $r = 0$, $C_{r,q,\rho,\beta,\eta}$ denotes the upper bound in (11) and if $\eta = 0$, $C_{r,q,\rho,\beta,\eta}$ denotes the upper bound in (12). Then we have the following convergence results:

**Theorem 1** *(strongly convex case) Let $F(\cdot)$ be $L$-smooth and $\mu$-strongly convex. With Assumption 1, Assumption 2, Assumption 3, and $\eta_t = \frac{1-\beta}{\mu(t+1)}$, $m = \lceil T/2 \rceil$, we obtain*

$$\frac{1}{m}\sum_{t=T-m}^{T-1}\mathbb{E}(F(\mathbf{w}_t) - F(\mathbf{w}^*)) \leq \frac{3A + 2G\sqrt{C_{(1-\beta)/\mu,1,1,\beta,0}}(1-\beta)}{\mu T}.$$

*where $A = \max\{\mu^2 D^2, 2(1-\beta)\sqrt{C_{(1-\beta)/\mu,1,1,\beta,0}}LD + \mu UD + 2G^2 + 2U^2\}$.*

**Theorem 2** *(convex case) Let $F(\cdot)$ be a convex function. With Assumption 1, Assumption 2, Assumption 3, and $\eta_t = \frac{1-\beta}{\sqrt{t+1}}$, we obtain*

$$\sum_{t=0}^{T-1}\frac{2}{\sqrt{t+1}}\mathbb{E}(F(\mathbf{w}_t) - F(\mathbf{w}^*)) \leq \|\mathbf{w}_0 - \mathbf{w}^*\|^2 + A\log(T),$$

*where $A = 2(1-\beta)\sqrt{C_{1-\beta,1,0.5,\beta,0}}G + 2UD + 2G^2 + 2U^2$.*

Since $\sum_{t=0}^{T-1}\frac{1}{\sqrt{t+1}} = \mathcal{O}(\sqrt{T})$, Theorem 2 implies that if the objective function is convex, GMC has a convergence rate of $\mathcal{O}(\log(T)/\sqrt{T})$.

**Theorem 3** *(non-convex case) Let $F(\cdot)$ be $L$-smooth. With Assumption 1, Assumption 2, and $\eta_t = \eta$ being a constant, we obtain*

$$\frac{1}{(1-\beta)T}\sum_{t=0}^{T-1}\mathbb{E}\|\nabla F(\mathbf{w}_t)\|^2 \leq \frac{F(\mathbf{w}_0) - F(\mathbf{w}^*)}{T\eta} + A\eta,$$

*where $A = \frac{LG\sqrt{C_{0,0,1,\beta,\eta}}}{1-\beta} + \frac{LG^2}{2(1-\beta)^2}$.*

By taking $\eta = \mathcal{O}(1/\sqrt{T})$, it is easy to find that GMC has a convergence rate of $\mathcal{O}(1/\sqrt{T})$, if the objective function is non-convex.

## 5 EXPERIMENTS

We conduct experiments on a PyTorch based parameter server framework with one server and eight workers. Each worker has access to one TITAN Xp GPU. We compare GMC with distributed momentum SGD (DMSGD) and DGC. In the experiments of (Lin et al., 2018), DGC gets far better performance on both accuracy and communication compression ratio than quantization methods. Hence, we do not compare with quantization methods in this paper. We use warm-up strategy (run DMSGD 5 epochs) for both GMC and DGC. The momentum factor masking trick is used in DGC. The $\beta$ is set as $0.9$. In our experiments, we consider the communication cost on the server which is the busiest node. It includes receiving vectors from the $p$ workers and sending one vector to the $p$ workers. So the cost of DMSGD is $2pd$. In GMC and DGC, since $\mathbf{m}_{t,k}$ is sparse, workers send the vectors using the structure of $(key, value)$. The cost of each $(key, value)$ is 2. Server sends $\mathbf{w}_{t+1} - \mathbf{w}_t$ using this structure as well. Hence, the cost of GMC and DGC is $2(\sum_{k=1}^{p} \|\mathbf{m}_{t,k}\|_0 + p\|\mathbf{w}_{t+1} - \mathbf{w}_t\|_0)$. We define the communication compression ratio (CR) of GMC and DGC as:

$$\text{CR} = \frac{1}{T} \sum_{t=5}^{T+4} \frac{1}{pd}(p\|\mathbf{w}_{t+1} - \mathbf{w}_t\|_0 + \sum_{k=1}^{p} \|\mathbf{m}_{t,k}\|_0).$$

The CR of DMSGD is $100\%$ (no compression). Here, all numbers have the same unit (float value).

### 5.1 CONVEX MODEL

We use the dataset MNIST and the model logistic regression (LR) to evaluate GMC on convex problems. Since the sizes of dataset and model are small, $\mathbf{m}_{t,k}$ directly adopts top-$s$ strategy with $\|\mathbf{m}_{t,k}\|_0 = s$ where $s = 0.01d$ or $0.001d$. We use 8 workers for this experiment. The weight decay is $0.0001$ and total batch size is 128. The result is shown in Figure 1. We can see that GMC gets similar performance as DGC and DMSGD.

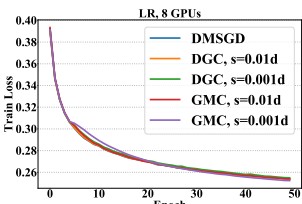 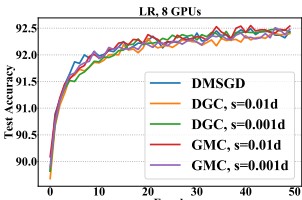

Figure 1: Comparison on convex model. The CRs of DGC with $s = 0.01d$ and $0.001d$ are $7.87\%$ and $0.797\%$ respectively. The CRs of GMC with $s = 0.01d$ and $0.001d$ are $8.0\%$ and $0.797\%$ respectively.

### 5.2 NON-CONVEX MODEL

For large models, we use the approximate top-$s$ strategy for $\mathbf{m}_{t,k}$: given a vector $\mathbf{a}$, we first randomly choose a subset $S \subset \{1, \ldots, d\}$ and $|S|$ is $0.001d \sim 0.01d$. We get the threshold $\theta$ such that $|\{j | |a^{(j)}| \geq \theta, j \in S\}| = 0.001|S|$. Then we set $\mathbf{m}_{t,k}$ by choosing the indexes in $\{j | |a^{(j)}| \geq \theta, j = 1, 2, \ldots, d\}$. It implies that $\|\mathbf{m}_{t,k}\|_0$ is approximately $0.001d$.

First, we use the dataset CIFAR-10 and three popular deep models (AlexNet, ResNet20, ResNet56) to evaluate GMC on non-convex problems. We use both 4 and 8 workers with the total batch size 128. The weight decay is set as $0.0001$. Figure 2 shows the learning process about training loss and test accuracy of the three methods. We can find that GMC and DGC have the same performance as that of DMSGD on ResNet20 and ResNet56. Compared to the two residual networks, AlexNet has more parameters. We can see that GMC also gets the performance as that of DMSGD and is slightly better than DGC. Table 1 shows the finial training results of the three methods. Compared with DMSGD, GMC can reduce the communication cost by more than 100 fold with no loss of accuracy. This is far better than those quantization methods. Furthermore, GMC achieves slightly better accuracy with comparable communication compression ratio, compared with DGC.

We also evaluate GMC on larger dataset ImageNet and larger model ResNet18. We use 8 workers with total batch size 256. The weight decay is set as $0.0001$. In Figure 3, GMC gets the same

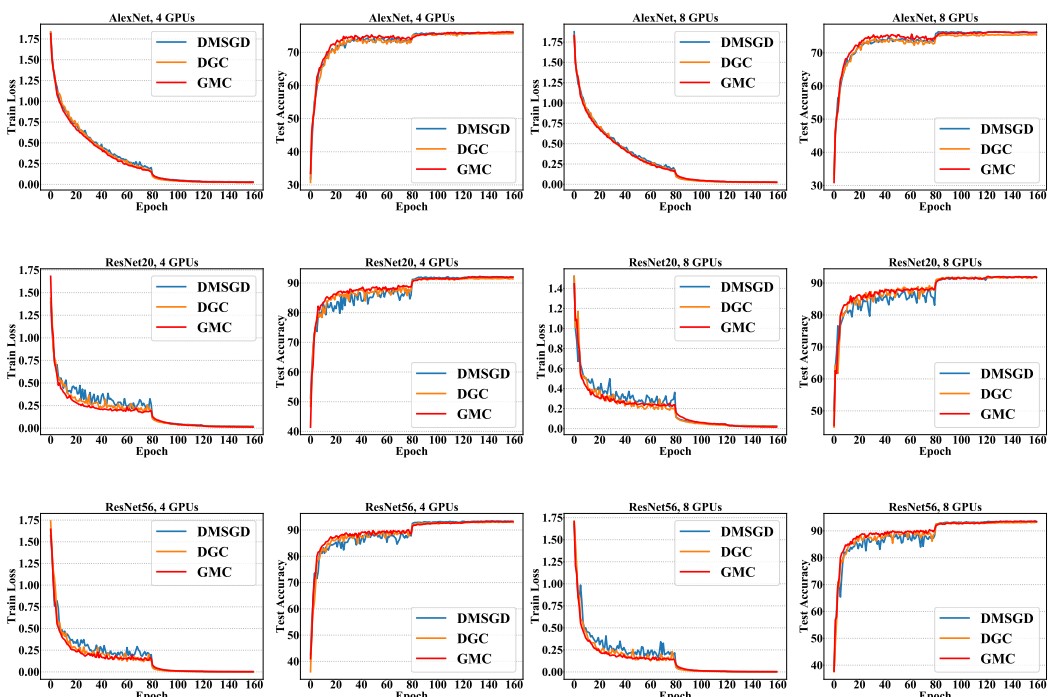

Figure 2: Learning process of using different non-convex models and workers on CIFAR10.

| GPU | Method | AlexNet | | ResNet20 | | ResNet56 | |
|---|---|---|---|---|---|---|---|
| | | Accuracy | CR | Accuracy | CR | Accuracy | CR |
| 4 | DMSGD | 75.76% | 100% | 91.93% | 100% | 93.23% | 100% |
| | DGC | 75.67% | 0.48% | 91.35% | 0.37% | 93.08% | 0.47% |
| | GMC | 76.10% | 0.48% | 91.97% | 0.36% | 93.23% | 0.46% |
| 8 | DMSGD | 76.08% | 100% | 91.79% | 100% | 93.47% | 100% |
| | DGC | 75.42% | 0.87% | 91.80% | 0.66% | 93.26% | 0.84% |
| | GMC | 76.19% | 0.85% | 91.98% | 0.64% | 93.43% | 0.81% |

Table 1: Experimental results on CIFAR10.

test accuracy as that of DMSGD during the learning process. Table 2 shows that compared with DMSGD, GMC can reduce the communication cost by more than 100 fold with no loss of accuracy and save about $40\%$ of training time. Compared with DGC, GMC achieves slightly better accuracy with comparable communication compress ratio.

| Method | Time (second/epoch) | Accuracy | CR |
|---|---|---|---|
| DMSGD | 2536 | 69.79% | 100% |
| DGC | 1496 | 69.57% | 0.84% |
| GMC | 1495 | 69.88% | 0.82% |

Table 2: Experimental results on ImageNet.

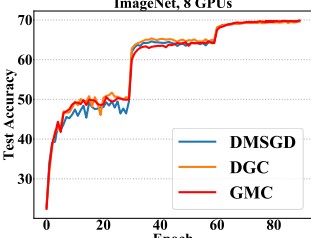

Figure 3: Learning process.

## 6 CONCLUSION

In this paper, we propose a novel method, called global momentum compression (GMC), for sparse communication in DMSGD (DSGD). To the best of our knowledge, this is the first work that proves the convergence of DMSGD with sparse communication and memory gradient. Empirical results show that GMC can achieve state-of-the-art performance.

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

# A    USING SPARSE COMMUNICATION ON SERVER

If we use the sparse communication technique on the server, the update for $\mathbf{w}_t$ can be written as:

$$\mathbf{w}_{t+1} = \mathbf{w}_t - \eta_t \mathbf{a}_t \odot [\sum_{k=1}^{p} \mathbf{m}_{t,k} \odot (\mathbf{g}_{t,k} + \mathbf{u}_{t,k}) + \Delta_t], \tag{13}$$

$$\mathbf{u}_{t+1,k} = (\mathbf{1} - \mathbf{m}_{t,k}) \odot (\mathbf{g}_{t,k} + \mathbf{u}_{t,k}), k = 1, 2, \ldots, p, \tag{14}$$

$$\Delta_{t+1} = (\mathbf{1} - \mathbf{a}_t) \odot [\sum_{k=1}^{p} \mathbf{m}_{t,k} \odot (\mathbf{g}_{t,k} + \mathbf{u}_{t,k}) + \Delta_t], \tag{15}$$

where $\mathbf{g}_{t,k}$ is the same as that in GMC. We call this method GMC$^*$. The $\mathbf{m}_{t,k}$ and $\mathbf{a}_t$ are generated by the worker and server respectively. The $\mathbf{u}_{t,k}$ and $\Delta_t$ are the memory gradients on the worker and server respectively. We can call $\mathbf{u}_{t,k}$ the local memory gradient and $\Delta_t$ the global memory gradient. Details are presented in Algorithm 2.

---

**Algorithm 2** GMC$^*$

---

1: Initialization: $p$ workers, $\mathbf{w}_{-1} = \mathbf{w}_0$, $\beta \in [0, 1)$, batch size $b$;
2: Set $\mathbf{g}_{0,k} = \mathbf{u}_{0,k} = \Delta_0 = \mathbf{0}, k = 1, \ldots, p$,
3: **for** $t = 0, 1, 2, \ldots T - 1$ **do**
4:      Workers:
5:      **for** $k = 1, 2 \ldots, p$, **each worker parallelly do**
6:          **if** $t > 0$ **then**
7:              Receive $\mathbf{w}_t - \mathbf{w}_{t-1}$ from the server;
8:              Get $\mathbf{w}_t$ by $\mathbf{w}_t = \mathbf{w}_{t-1} + (\mathbf{w}_t - \mathbf{w}_{t-1})$;
9:          **end if**
10:         Randomly pick a mini-batch of training data $\mathcal{I}_{t,k} \subseteq D_k$ with $|\mathcal{I}_{t,k}| = b$;
11:         $\mathbf{g}_{t,k} = \frac{1}{pb} \sum_{\xi_i \in \mathcal{I}_{t,k}} \nabla f(\mathbf{w}_t; \xi_i) - \frac{\beta}{p\eta_t}(\mathbf{w}_t - \mathbf{w}_{t-1})$;
12:         Generate a sparse vector $\mathbf{m}_{t,k} \in \{0, 1\}^d$;
13:         Send $\mathbf{m}_{t,k} \odot (\mathbf{g}_{t,k} + \mathbf{u}_{t,k})$ to the server;
14:         $\mathbf{u}_{t+1,k} = (\mathbf{1} - \mathbf{m}_{t,k}) \odot (\mathbf{g}_{t,k} + \mathbf{u}_{t,k})$;
15:     **end for**
16:     Server:
17:     Generate a sparse vector $\mathbf{a}_t \in \{0, 1\}^d$;
18:     $\mathbf{w}_{t+1} = \mathbf{w}_t - \eta_t \mathbf{a}_t \odot [\sum_{k=1}^{p} \mathbf{m}_{t,k} \odot (\mathbf{g}_{t,k} + \mathbf{u}_{t,k}) + \Delta_t]$;
19:     $\Delta_{t+1} = (\mathbf{1} - \mathbf{a}_t) \odot [\sum_{k=1}^{p} \mathbf{m}_{t,k} \odot (\mathbf{g}_{t,k} + \mathbf{u}_{t,k}) + \Delta_t]$
20:     Send $\mathbf{w}_{t+1} - \mathbf{w}_t$ to workers;
21: **end for**

---

We eliminate $\mathbf{m}_{t,k}, \mathbf{a}_t$ and obtain

$$\mathbf{w}_{t+1} - \eta_t \Delta_{t+1} = \mathbf{w}_t - \eta_t [\sum_{k=1}^{p} \mathbf{m}_{t,k} \odot (\mathbf{g}_{t,k} + \mathbf{u}_{t,k}) + \Delta_t]$$

$$= \mathbf{w}_t - \eta_t [\tilde{\mathbf{g}}_t + \tilde{\mathbf{u}}_t - \tilde{\mathbf{u}}_{t+1} + \Delta_t]$$

By denoting $\mathbf{h}_t = \tilde{\mathbf{u}}_t + \Delta_t$, we obtain

$$\mathbf{w}_{t+1} - \eta_t \mathbf{h}_{t+1} = \mathbf{w}_t - \eta_t \mathbf{h}_t - \eta_t \nabla f(\mathbf{w}_t; \mathcal{I}_t) + \beta(\mathbf{w}_t - \mathbf{w}_{t-1})$$

The equation is the same as (9). Hence our convergence analysis is also suitable for GMC$^*$. In Figure 4, we set $\mathbf{a}_t$ using top-$s$ strategy with $\|\mathbf{a}_t\|_0 = s = 0.001d$, which implies that $\mathbf{w}_{t+1} - \mathbf{w}_t$ is nearly as sparse as $\mathbf{m}_{t,k}$. We can see that GMC$^*$ achieves the same loss as DMSGD. But it is slightly worse than GMC and DMSGD in terms of accuracy.

# B    OTHER WAYS FOR SPARSE COMMUNICATION

It is easy to get one way to combine memory gradient and global momentum that we can put the global momentum $\mathbf{w}_t - \mathbf{w}_{t-1}$ on the server. We briefly present it in Algorithm 3.

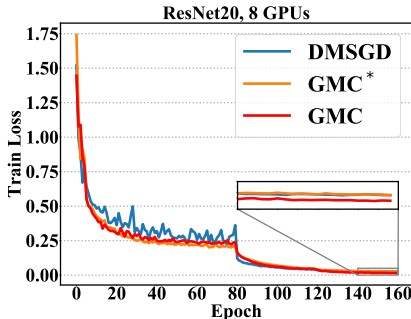 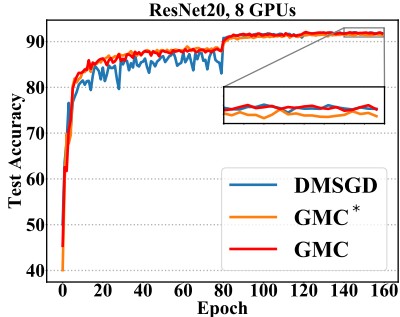

Figure 4: Comparing GMC* with GMC and DMSGD.

---

**Algorithm 3**

---

**for** $t = 0, 1, 2, \ldots T - 1$ **do**
  Workers:
  **for** $k = 1, 2 \ldots, p$, **each worker parallelly do**
    $\mathbf{g}_{t,k} = \frac{1}{pb} \sum_{\xi_i \in \mathcal{I}_{t,k}} \nabla f(\mathbf{w}_t; \xi_i)$;
    Generate a sparse vector $\mathbf{m}_{t,k} \in \{0,1\}^d$;
    Send $\mathbf{m}_{t,k} \odot (\mathbf{g}_{t,k} + \mathbf{u}_{t,k})$ to the server;
    $\mathbf{u}_{t+1,k} = (\mathbf{1} - \mathbf{m}_{t,k}) \odot (\mathbf{g}_{t,k} + \mathbf{u}_{t,k})$;
  **end for**
  Server:
  $\mathbf{w}_{t+1} = \mathbf{w}_t - \eta_t \sum_{k=1}^{p} \mathbf{m}_{t,k} \odot (\mathbf{g}_{t,k} + \mathbf{u}_{t,k}) + \beta(\mathbf{w}_t - \mathbf{w}_{t-1})$;
  Send $\mathbf{w}_{t+1} - \mathbf{w}_t$ to workers;
**end for**

---

Let $\{\mathbf{w}_t\}$, $\{\mathbf{g}_{t,k}\}$, $\{\mathbf{u}_{t,k}\}$ be the sequences produced by Algorithm 3, then we can get that

$$\mathbf{w}_{t+1} = \mathbf{w}_t - \eta_t \sum_{k=1}^{p} \mathbf{M}_{t,k}(\mathbf{g}_{t,k} + \mathbf{u}_{t,k}) + \beta(\mathbf{w}_t - \mathbf{w}_{t-1}),$$

$$\mathbf{u}_{t+1,k} = (\mathbf{I} - \mathbf{M}_{t,k})(\mathbf{g}_{t,k} + \mathbf{u}_{t,k}), k = 1, 2, \ldots, p.$$

By eliminating $\mathbf{M}_{t,k}$, we obtain the same equation as that in (9). Hence our convergence analysis is also suitable for Algorithm 3. The difference from GMC is that the memory gradient $\mathbf{u}_{t,k}$ in Algorithm 3 only remembers the dropped stochastic gradient. After receiving sparse update vectors from workers, server updates parameter using the global momentum. Compared with GMC, the disadvantage is that its memory gradient does not contain the momentum information which can play the role of correcting the update direction. Hence, GMC gets better performance than Algorithm 3. (Lin et al., 2018) also points out that Algorithm 3 has some loss of convergence performance.

Another way is that we put $\eta_t$ in the memory gradient. We briefly present it in Algorithm 4.

Let $\{\mathbf{w}_t\}$, $\{\mathbf{g}_{t,k}\}$, $\{\mathbf{u}_{t,k}\}$ be the sequences produced by Algorithm 4, then we can get that

$$\mathbf{w}_{t+1} = \mathbf{w}_t - \sum_{k=1}^{p} \mathbf{M}_{t,k}(\mathbf{g}_{t,k} + \mathbf{u}_{t,k})$$

$$\mathbf{u}_{t+1,k} = (\mathbf{I} - \mathbf{M}_{t,k})(\mathbf{g}_{t,k} + \mathbf{u}_{t,k}), k = 1, 2, \ldots, p.$$

By eliminating $\mathbf{M}_{t,k}$, we obtain the same equation as that in (9). Hence our convergence analysis is also suitable for Algorithm 3. However, the norm of memory gradient is larger than that of GMC. In particular, let $\mathbf{y}_{t,k} = \mathbf{u}_{t,k}/\eta_t$, then we have

$$\mathbf{y}_{t+1,k} = \frac{\eta_t}{\eta_{t+1}} \underbrace{(\mathbf{I} - \mathbf{M}_{t,k})(\frac{1}{pb} \sum_{\xi_i \in \mathcal{I}_{t,k}} \nabla f(\mathbf{w}_t; \xi_i) - \frac{\beta}{p\eta_t}(\mathbf{w}_t - \mathbf{w}_{t-1}) + \mathbf{y}_{t,k})}_{\text{the same as the memory gradient of GMC}}$$

---

**Algorithm 4**

> **for** $t = 0, 1, 2, \ldots T-1$ **do**
>> Workers:
>> **for** $k = 1, 2 \ldots, p$, **each worker parallelly do**
>>> $\mathbf{g}_{t,k} = \frac{\eta_t}{pb} \sum_{\xi_i \in \mathcal{I}_{t,k}} \nabla f(\mathbf{w}_t; \xi_i) - \frac{\beta}{p}(\mathbf{w}_t - \mathbf{w}_{t-1})$;
>>> Generate a sparse vector $\mathbf{m}_{t,k} \in \{0,1\}^d$;
>>> Send $\mathbf{m}_{t,k} \odot (\mathbf{g}_{t,k} + \mathbf{u}_{t,k})$ to the server;
>>> $\mathbf{u}_{t+1,k} = (\mathbf{1} - \mathbf{m}_{t,k}) \odot (\mathbf{g}_{t,k} + \mathbf{u}_{t,k})$;
>> **end for**
>> Server:
>> $\mathbf{w}_{t+1} = \mathbf{w}_t - \sum_{k=1}^{p} \mathbf{m}_{t,k} \odot (\mathbf{g}_{t,k} + \mathbf{u}_{t,k})$;
>> Send $\mathbf{w}_{t+1} - \mathbf{w}_t$ to workers;
> **end for**

---

If $\eta_t$ is a constant, then Algorithm 4 is the same as GMC. If $\eta_t$ is non-increasing, then $\eta_t \geq \eta_{t+1}$ and

$$\|\mathbf{y}_{t+1,k}\| \geq \|(\mathbf{I} - \mathbf{M}_{t,k})(\frac{1}{pb}\sum_{\xi_i \in \mathcal{I}_{t,k}} \nabla f(\mathbf{w}_t; \xi_i) - \frac{\beta}{p\eta_t}(\mathbf{w}_t - \mathbf{w}_{t-1}) + \mathbf{y}_{t,k})\|.$$

Figure 5 shows the learning process of ResNet56 on CIFAR10. In the first $80$ epochs, GMC and Algorithm 4 are almost the same and both of them are better than Algorithm 3. At the step of scaling step size by factor $0.1$, the norm of memory gradient ($\mathbf{y}_{t,k}$) in Algorithm 4 is 10 fold of that in GMC. Hence, GMC is better than Algorithm 4 after training $80$ epochs.

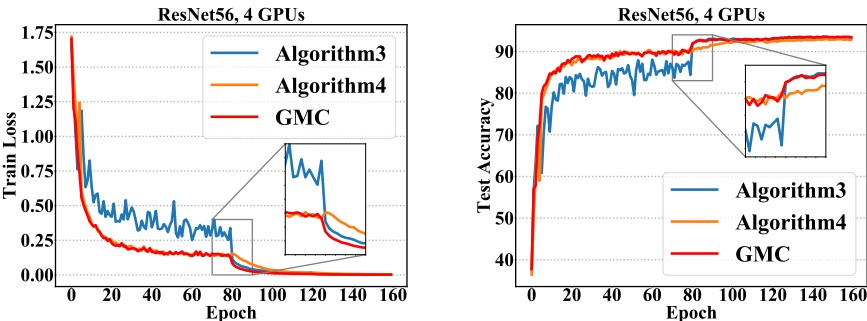

Figure 5: Comparison with Algorithm 3 and Algorithm 4.

## C  CONVERGENCE PROOFS

First, we introduce the Young's Inequality as follow: $\forall a, b$

$$(a+b)^2 \leq (1+\gamma)a^2 + (1+\frac{1}{\gamma})b^2, \forall \gamma > 0,$$

which is used frequently in our proofs.

## C.1 PROOF OF LEMMA 1

$$
\begin{aligned}
\mathbf{x}_{t+1} =& \mathbf{w}_{t+1} + \frac{\beta}{1-\beta}(\mathbf{w}_{t+1} - \mathbf{w}_t) - \frac{\eta_{t+1}}{1-\beta}\tilde{\mathbf{u}}_{t+1} \\
=& \frac{1}{1-\beta}(\mathbf{w}_{t+1} - \beta\mathbf{w}_t) - \frac{\eta_{t+1}}{1-\beta}\tilde{\mathbf{u}}_{t+1} \\
=& \frac{1}{1-\beta}(\mathbf{w}_t - \eta_t\tilde{\mathbf{u}}_t - \eta_t\nabla f(\mathbf{w}_t;\mathcal{I}_t) + \beta(\mathbf{w}_t - \mathbf{w}_{t-1}) - \beta\mathbf{w}_t + \eta_t\tilde{\mathbf{u}}_{t+1}) - \frac{\eta_{t+1}}{1-\beta}\tilde{\mathbf{u}}_{t+1} \\
=& \frac{1}{1-\beta}(\mathbf{w}_t - \beta\mathbf{w}_{t-1} - \eta_t\tilde{\mathbf{u}}_t - \eta_t\nabla f(\mathbf{w}_t;\mathcal{I}_t) + \eta_t\tilde{\mathbf{u}}_{t+1}) - \frac{\eta_{t+1}}{1-\beta}\tilde{\mathbf{u}}_{t+1} \\
=& \frac{1}{1-\beta}(\mathbf{w}_t - \beta\mathbf{w}_{t-1} - \eta_t\tilde{\mathbf{u}}_t - \eta_t\nabla f(\mathbf{w}_t;\mathcal{I}_t)) + \frac{\eta_t - \eta_{t+1}}{1-\beta}\tilde{\mathbf{u}}_{t+1} \\
=& \frac{1}{1-\beta}(\mathbf{w}_t - \beta\mathbf{w}_{t-1}) - \frac{\eta_t}{1-\beta}\tilde{\mathbf{u}}_t - \frac{\eta_t}{1-\beta}\nabla f(\mathbf{w}_t;\mathcal{I}_t) + \frac{\eta_t - \eta_{t+1}}{1-\beta}\tilde{\mathbf{u}}_{t+1} \\
=& \mathbf{x}_t - \frac{\eta_t}{1-\beta}\nabla f(\mathbf{w}_t;\mathcal{I}_t) + \frac{\eta_t - \eta_{t+1}}{1-\beta}\tilde{\mathbf{u}}_{t+1}.
\end{aligned}
$$

## C.2 PROOF OF LEMMA 2

Since $\mathbf{w}_{t+1} - \mathbf{w}_t = \eta_t \sum_{k=1}^p \mathbf{M}_{t,k}(\mathbf{g}_{t,k} + \mathbf{u}_{t,k})$, we have $\forall \gamma > 0$,

$$
\begin{aligned}
\mathbb{E}\|\mathbf{w}_{t+1} - \mathbf{w}_t\|^2 \leq& \eta_t^2\mathbb{E}\|\tilde{\mathbf{g}}_t + \tilde{\mathbf{u}}_t\|^2 \\
=& \eta_t^2\mathbb{E}\|\beta(\mathbf{w}_{t-1} - \mathbf{w}_t)/\eta_t + \nabla f(\mathbf{w}_t;\mathcal{I}_t) + \tilde{\mathbf{u}}_t\|^2 \\
\leq& (1+\gamma)\beta^2\mathbb{E}\|\mathbf{w}_t - \mathbf{w}_{t-1}\|^2 + (1+\frac{1}{\gamma})(2G^2 + 2U^2)\eta_t^2.
\end{aligned}
$$

Let $\gamma = \frac{1}{\beta} - 1$, we get that

$$
\begin{aligned}
\mathbb{E}\|\mathbf{w}_{t+1} - \mathbf{w}_t\|^2 \leq& \beta\mathbb{E}\|\mathbf{w}_t - \mathbf{w}_{t-1}\|^2 + \frac{\eta_t^2}{1-\beta}(2G^2 + 2U^2) \\
\leq& \frac{2G^2 + 2U^2}{1-\beta}(\eta_t^2 + \beta\eta_{t-1}^2 + \ldots + \beta^t\eta_0^2).
\end{aligned}
$$

If $\eta_t = \eta$, we get that $\mathbb{E}\|\mathbf{w}_t - \mathbf{w}_{t-1}\|^2 \leq \frac{(2G^2+2U^2)\eta^2}{(1-\beta)^2}$. Then we obtain

$$
\begin{aligned}
\mathbb{E}\|\mathbf{x}_t - \mathbf{w}_t\|^2 \leq& \frac{2\beta^2}{(1-\beta)^2}\mathbb{E}\|\mathbf{w}_t - \mathbf{w}_{t-1}\|^2 + \frac{2\eta^2}{(1-\beta)^2}U^2 \\
\leq& [\frac{2\beta^2(2G^2 + 2U^2)}{(1-\beta)^4} + \frac{2U^2}{(1-\beta)^2}]\eta^2.
\end{aligned}
$$

If $\eta_t = \frac{r}{(t+q)^\rho}, \rho \in [0.5, 1], r > 0, q \geq 0$, let $t_0$ satisfy $(\frac{t_0+1+q}{t_0+2+q})^{2\rho} \geq \frac{1+\beta}{2}$, $A = \max_{t=1}^{t_0}\{\frac{\eta_t^2+\beta\eta_{t-1}^2+\ldots+\beta^t\eta_0^2}{\eta_{t+1}^2}\}$. Then by induction, we get that $\mathbb{E}\|\mathbf{w}_t - \mathbf{w}_{t-1}\|^2 \leq \frac{(2G^2+2U^2)\max\{(1-\beta)A,2\}\eta_t^2}{(1-\beta)^2}$. Then we obtain

$$
\begin{aligned}
\mathbb{E}\|\mathbf{x}_t - \mathbf{w}_t\|^2 \leq& \frac{2\beta^2}{(1-\beta)^2}\mathbb{E}\|\mathbf{w}_t - \mathbf{w}_{t-1}\|^2 + \frac{2\eta_t^2}{(1-\beta)^2}U^2 \\
\leq& [\frac{2\beta^2(2G^2 + 2U^2)\max\{(1-\beta)A, 2\}}{(1-\beta)^4} + \frac{2U^2}{(1-\beta)^2}]\eta_t^2.
\end{aligned}
$$

## C.3 PROOF OF THEOREM 1

According to Lemma 1, we get that

$$
\begin{aligned}
&\mathbb{E}\|\mathbf{x}_{t+1} - \mathbf{w}^*\|^2 \\
=&\mathbb{E}\|\mathbf{x}_t - \mathbf{w}^*\|^2 - 2\mathbb{E}[\frac{\eta_t}{1-\beta}\nabla f(\mathbf{w}_t; \mathcal{I}_t) - \frac{\eta_t - \eta_{t+1}}{1-\beta}\tilde{\mathbf{u}}_{t+1}]^T(\mathbf{x}_t - \mathbf{w}^*) \\
&+ \mathbb{E}\|\frac{\eta_t}{1-\beta}\nabla f(\mathbf{w}_t; \mathcal{I}_t) - \frac{\eta_t - \eta_{t+1}}{1-\beta}\tilde{\mathbf{u}}_{t+1}\|^2 \\
=&\mathbb{E}\|\mathbf{x}_t - \mathbf{w}^*\|^2 - 2\mathbb{E}[\frac{\eta_t}{1-\beta}\nabla F(\mathbf{w}_t) - \frac{\eta_t - \eta_{t+1}}{1-\beta}\tilde{\mathbf{u}}_{t+1}]^T(\mathbf{x}_t - \mathbf{w}^*) \\
&+ \mathbb{E}\|\frac{\eta_t}{1-\beta}\nabla f(\mathbf{w}_t; \mathcal{I}_t) - \frac{\eta_t - \eta_{t+1}}{1-\beta}\tilde{\mathbf{u}}_{t+1}\|^2 \\
\leq&\mathbb{E}\|\mathbf{x}_t - \mathbf{w}^*\|^2 - \frac{2\eta_t}{1-\beta}\mathbb{E}\nabla F(\mathbf{w}_t)^T(\mathbf{x}_t - \mathbf{w}^*) + \frac{2(\eta_t - \eta_{t+1})}{1-\beta}\mathbb{E}\tilde{\mathbf{u}}_{t+1}^T(\mathbf{x}_t - \mathbf{w}^*) \\
&+ \mathbb{E}\|\frac{\eta_t}{1-\beta}\nabla f(\mathbf{w}_t; \mathcal{I}_t) - \frac{\eta_t - \eta_{t+1}}{1-\beta}\tilde{\mathbf{u}}_{t+1}\|^2 \\
\leq&\mathbb{E}\|\mathbf{x}_t - \mathbf{w}^*\|^2 - \frac{2\eta_t}{1-\beta}\mathbb{E}\nabla F(\mathbf{x}_t)^T(\mathbf{x}_t - \mathbf{w}^*) + \frac{2\eta_t}{1-\beta}\mathbb{E}(\nabla F(\mathbf{x}_t) - \nabla F(\mathbf{w}_t))^T(\mathbf{x}_t - \mathbf{w}^*) \\
&+ \frac{2(\eta_t - \eta_{t+1})}{1-\beta}UD + \frac{2G^2\eta_t^2}{(1-\beta)^2} + \frac{2(\eta_t - \eta_{t+1})^2}{(1-\beta)^2}U^2 \\
\leq&\mathbb{E}\|\mathbf{x}_t - \mathbf{w}^*\|^2 - \frac{2\eta_t}{1-\beta}\mathbb{E}F(\mathbf{x}_t)^T(\mathbf{x}_t - \mathbf{w}^*) + \frac{2\eta_t^2}{1-\beta}\sqrt{C_{(1-\beta)/\mu,0,1,\beta,0}}LD \\
&+ \frac{2(\eta_t - \eta_{t+1})}{1-\beta}UD + \frac{2G^2\eta_t^2}{(1-\beta)^2} + \frac{2(\eta_t - \eta_{t+1})^2}{(1-\beta)^2}U^2.
\end{aligned}
\tag{16}
$$

Using the strongly convex property, we get that

$$
\begin{aligned}
&\mathbb{E}\|\mathbf{x}_{t+1} - \mathbf{w}^*\|^2 \\
\leq&(1 - \frac{2}{t})\mathbb{E}\|\mathbf{x}_t - \mathbf{w}^*\|^2 + \frac{2\eta_t^2}{1-\beta}\sqrt{C_{(1-\beta)/\mu,0,1,\beta,0}}LD \\
&+ \frac{2(\eta_t - \eta_{t+1})}{1-\beta}UD + \frac{2G^2\eta_t^2}{(1-\beta)^2} + \frac{2(\eta_t - \eta_{t+1})^2}{(1-\beta)^2}U^2 \\
\leq&(1 - \frac{2}{t})\mathbb{E}\|\mathbf{x}_t - \mathbf{w}^*\|^2 + [2(1-\beta)\sqrt{C_{(1-\beta)/\mu,0,1,\beta,0}}LD + 2\mu UD + 2G^2 + 2U^2]\frac{1}{\mu^2 t^2}.
\end{aligned}
$$

Since $\mathbb{E}\|\mathbf{x}_1 - \mathbf{w}^*\|^2 \leq D^2$, we get that

$$
\mathbb{E}\|\mathbf{x}_t - \mathbf{w}^*\|^2 \leq \frac{A}{\mu^2 t}, \forall t \geq 1,
$$

where $A = \max\{\mu^2 D^2, 2(1-\beta)\sqrt{C_{(1-\beta)/\mu,0,1,\beta,0}}LD + 2\mu UD + 2G^2 + 2U^2\}$. Using (16) again, we get that

$$
2\mathbb{E}(F(\mathbf{x}_t) - F(\mathbf{w}^*)) \leq \mu t(\mathbb{E}\|\mathbf{x}_t - \mathbf{w}^*\|^2 - \mathbb{E}\|\mathbf{x}_{t+1} - \mathbf{w}^*\|^2) + \frac{A}{\mu t}.
$$

By summing up the equation from $t = T - m$ to $T - 1$, we get that

$$2 \sum_{t=T-m}^{T-1} \mathbb{E}(F(\mathbf{x}_t) - F(\mathbf{w}^*))$$

$$\leq \sum_{t=T-m}^{T-1} [\mu t(\mathbb{E}\|\mathbf{x}_t - \mathbf{w}^*\|^2 - \mathbb{E}\|\mathbf{x}_{t+1} - \mathbf{w}^*\|^2) + \frac{A}{\mu t}]$$

$$\leq \mu(T - m)\mathbb{E}\|\mathbf{x}_{T-m} - \mathbf{w}^*\|^2 + \sum_{t=T-m}^{T-1} (\mu\mathbb{E}\|\mathbf{x}_t - \mathbf{w}^*\|^2 + \frac{A}{\mu t})$$

$$\leq \mu(T - m)\mathbb{E}\|\mathbf{x}_{T-m} - \mathbf{w}^*\|^2 + \sum_{t=T-m}^{T-1} \frac{2A}{\mu t}$$

$$\leq \frac{A}{\mu} + \sum_{t=T-m}^{T-1} \frac{2A}{\mu t}.$$

By setting $m = \lceil T/2 \rceil$, we get that $\sum_{t=T-m}^{T-1} \frac{1}{t} \leq 1$. Then

$$\frac{1}{m} \sum_{t=T-m}^{T-1} \mathbb{E}(F(\mathbf{x}_t) - F(\mathbf{w}^*)) \leq \frac{3A}{\mu T}.$$

Since $\mathbb{E}[F(\mathbf{x}_t) - F(\mathbf{w}_t)] \geq \mathbb{E}\nabla F(\mathbf{w}_t)^T(\mathbf{x}_t - \mathbf{w}_t) \geq -G\sqrt{C_{(1-\beta)/\mu,0,1,\beta,0}}\frac{1-\beta}{\mu t}$, we get the result

$$\frac{1}{m} \sum_{t=T-m}^{T-1} \mathbb{E}(F(\mathbf{w}_t) - F(\mathbf{w}^*)) \leq \frac{3A + 2G\sqrt{C_{(1-\beta)/\mu,0,1,\beta,0}}(1 - \beta)}{\mu T}.$$

## C.4 Proof of Theorem 2

Similar to (16), we get that

$$\mathbb{E}\|\mathbf{x}_{t+1} - \mathbf{w}^*\|^2$$

$$= \mathbb{E}\|\mathbf{x}_t - \mathbf{w}^*\|^2 - 2\mathbb{E}[\frac{\eta_t}{1-\beta}\nabla f(\mathbf{w}_t; \mathcal{I}_t) - \frac{\eta_t - \eta_{t+1}}{1-\beta}\tilde{\mathbf{u}}_{t+1}]^T(\mathbf{x}_t - \mathbf{w}^*)$$

$$+ \mathbb{E}\|\frac{\eta_t}{1-\beta}\nabla f(\mathbf{w}_t; \mathcal{I}_t) - \frac{\eta_t - \eta_{t+1}}{1-\beta}\tilde{\mathbf{u}}_{t+1}\|^2$$

$$= \mathbb{E}\|\mathbf{x}_t - \mathbf{w}^*\|^2 - 2\mathbb{E}[\frac{\eta_t}{1-\beta}\nabla F(\mathbf{w}_t) - \frac{\eta_t - \eta_{t+1}}{1-\beta}\tilde{\mathbf{u}}_{t+1}]^T(\mathbf{x}_t - \mathbf{w}^*)$$

$$+ \mathbb{E}\|\frac{\eta_t}{1-\beta}\nabla f(\mathbf{w}_t; \mathcal{I}_t) - \frac{\eta_t - \eta_{t+1}}{1-\beta}\tilde{\mathbf{u}}_{t+1}\|^2$$

$$\leq \mathbb{E}\|\mathbf{x}_t - \mathbf{w}^*\|^2 - \frac{2\eta_t}{1-\beta}\mathbb{E}\nabla F(\mathbf{w}_t)^T(\mathbf{x}_t - \mathbf{w}^*) + \frac{2(\eta_t - \eta_{t+1})}{1-\beta}\mathbb{E}\tilde{\mathbf{u}}_{t+1}^T(\mathbf{x}_t - \mathbf{w}^*)$$

$$+ \mathbb{E}\|\frac{\eta_t}{1-\beta}\nabla f(\mathbf{w}_t; \mathcal{I}_t) - \frac{\eta_t - \eta_{t+1}}{1-\beta}\tilde{\mathbf{u}}_{t+1}\|^2$$

$$\leq \mathbb{E}\|\mathbf{x}_t - \mathbf{w}^*\|^2 - \frac{2\eta_t}{1-\beta}\mathbb{E}\nabla F(\mathbf{w}_t)^T(\mathbf{w}_t - \mathbf{w}^*) + \frac{2\eta_t}{1-\beta}\mathbb{E}\nabla F(\mathbf{w}_t)^T(\mathbf{w}_t - \mathbf{x}_t)$$

$$+ \frac{2(\eta_t - \eta_{t+1})}{1-\beta}UD + \frac{2G^2\eta_t^2}{(1-\beta)^2} + \frac{2(\eta_t - \eta_{t+1})^2}{(1-\beta)^2}U^2$$

$$\leq \mathbb{E}\|\mathbf{x}_t - \mathbf{w}^*\|^2 - \frac{2\eta_t}{1-\beta}\mathbb{E}(F(\mathbf{w}_t) - F(\mathbf{w}^*)) + \frac{2\eta_t^2}{1-\beta}\sqrt{C_{1-\beta,1,0.5,\beta,0}}G$$

$$+ \frac{2(\eta_t - \eta_{t+1})}{1-\beta}UD + \frac{2G^2\eta_t^2}{(1-\beta)^2} + \frac{2(\eta_t - \eta_{t+1})^2}{(1-\beta)^2}U^2. \tag{17}$$

Since $\eta_t = \frac{1-\beta}{\sqrt{t+1}}$, we get that

$$\sum_{t=0}^{T-1} \frac{2}{\sqrt{t+1}} \mathbb{E}(F(\mathbf{w}_t) - F(\mathbf{w}^*)) \le \|\mathbf{w}_0 - \mathbf{w}^*\|^2 + \sum_{t=0}^{T-1} \frac{A}{t+1}$$

$$\le \|\mathbf{w}_0 - \mathbf{w}^*\|^2 + A\log(T),$$

where $A = 2(1-\beta)\sqrt{C_{1-\beta,1,0.5,\beta,0}}G + 2UD + 2G^2 + 2U^2$.

### C.5 PROOF OF THEOREM 3

Since $F(\cdot)$ is $L$-smooth, we get that

$$\mathbb{E}F(\mathbf{x}_{t+1})$$

$$\le \mathbb{E}F(\mathbf{x}_t) - \frac{\eta}{1-\beta} \mathbb{E}[\nabla F(\mathbf{x}_t)^T \nabla f(\mathbf{w}_t; \mathcal{I}_t)] + \frac{L}{2}\mathbb{E}\|\frac{\eta}{1-\beta}\nabla f(\mathbf{w}_t; \mathcal{I}_t)\|^2$$

$$= \mathbb{E}F(\mathbf{x}_t) - \frac{\eta}{1-\beta} \mathbb{E}[\nabla F(\mathbf{x}_t)^T \nabla F(\mathbf{w}_t)] + \frac{L\eta^2}{2(1-\beta)^2} \mathbb{E}\|\nabla f(\mathbf{w}_t; \mathcal{I}_t)\|^2$$

$$\le \mathbb{E}F(\mathbf{x}_t) - \frac{\eta}{1-\beta} \mathbb{E}[\|\nabla F(\mathbf{w}_t)\|^2 + (\nabla F(\mathbf{x}_t) - \nabla F(\mathbf{w}_t))^T \nabla F(\mathbf{w}_t)] + \frac{LG^2\eta^2}{2(1-\beta)^2}$$

$$\le \mathbb{E}F(\mathbf{x}_t) - \frac{\eta}{1-\beta} \mathbb{E}[\|\nabla F(\mathbf{w}_t)\|^2 - LG\|\mathbf{x}_t - \mathbf{w}_t\|] + \frac{LG^2\eta^2}{2(1-\beta)^2}$$

$$\le \mathbb{E}F(\mathbf{x}_t) - \frac{\eta}{1-\beta} \mathbb{E}\|\nabla F(\mathbf{w}_t)\|^2 + \frac{LG\sqrt{C_{0,0,1,\beta,\eta}}\eta^2}{1-\beta} + \frac{LG^2\eta^2}{2(1-\beta)^2}$$

Summing up the above equation from $t = 0$ to $T - 1$, we get that

$$\frac{1}{(1-\beta)T} \sum_{t=0}^{T-1} \mathbb{E}\|\nabla F(\mathbf{w}_t)\|^2 \le \frac{F(\mathbf{w}_0) - F(\mathbf{w}^*)}{T\eta} + A\eta,$$

where $A = \frac{LG\sqrt{C_{0,0,1,\beta,\eta}}}{1-\beta} + \frac{LG^2}{2(1-\beta)^2}$.

## D MORE DISCUSSION ABOUT ASSUMPTION 2

In the convergence theorems, we need $\mathbb{E}\|\tilde{\mathbf{u}}_t\|^2 \le U^2$ (Assumption 2). Since $\tilde{\mathbf{u}}_t = \sum_{k=1}^p \mathbf{u}_{t,k}$, we only need $\mathbb{E}\|\mathbf{u}_{t,k}\|^2$ to be bounded which is mainly related to the choice of $\mathbf{m}_{t,k}$. Theoretically, we set one large threshold $\tilde{\theta} > 0$ in advance and we denote $\theta_{t,k}^s$ as the $s$-largest value of $\{|g_{t,k}^{(j)} + u_{t,k}^{(j)}| | j = 1, \ldots, d\}$. Then in each iteration, we can choose the values such that $|g_{t,k}^{(j)} + u_{t,k}^{(j)}| \ge \min\{\tilde{\theta}, \theta_{t,k}^s\}$. Then we obtain $\|\mathbf{u}_{t,k}\|^2 \le d\tilde{\theta}^2$ and hence $\mathbb{E}\|\mathbf{u}_{t,k}\|^2 \le d\tilde{\theta}^2$. In practice, the large threshold $\tilde{\theta}$ is never been used for the choice of $\mathbf{m}_{t,k}$ since $\theta_{t,k}^s \ll \tilde{\theta}$.

## E MORE DISCUSSION ABOUT ASSUMPTION 3

If we do not use Assumption 3 in the proofs of Theorem 1 and Theorem 2, we can add one projection operation on equation (7), which can be written as

$$\mathbf{w}_{t+1} = \Pi_\Omega(\mathbf{w}_t - \eta_t \sum_{k=1}^p \mathbf{M}_{t,k}(\mathbf{g}_{t,k} + \mathbf{u}_{t,k})),$$

where $\Omega = \{\mathbf{w} | \|\mathbf{w}\| \le W\}$ and $\|\mathbf{w}^*\| \le \frac{W}{2}$. Then $\mathbf{w}_t$ is bounded. Hence, we reasonably assume that $\forall t, i, \|\tilde{\mathbf{u}}_t\| \le U, \|\nabla f(\mathbf{w}_t; \xi_i)\| \le G$ and we set $\eta_t \le \eta_0 \le \frac{(1-\beta)W}{2U}$. Please note that the $U$ can be irrelevant to $\Omega$ according to Appendix D. We still define $\mathbf{x}_t$ as

$$\mathbf{x}_t = \mathbf{w}_t + \frac{\beta}{1-\beta}(\mathbf{w}_t - \mathbf{w}_{t-1}) - \frac{\eta_t}{1-\beta}\tilde{\mathbf{u}}_t.$$

Then $\mathbf{x}_t$ is bounded as well.

To get the convergence results for strongly convex and convex functions, we need to prove two essential inequalities: $\mathbb{E}\|\mathbf{x}_t - \mathbf{w}_t\|^2 \leq \mathcal{O}(\eta_t^2)$ and $\|\mathbf{x}_{t+1} - \mathbf{w}^*\|^2 \leq \|\mathbf{x}_t - \mathbf{w}^* - \frac{\eta_t}{1-\beta}\nabla f(\mathbf{w}_t; \mathcal{I}_t) + \frac{\eta_t - \eta_{t+1}}{1-\beta}\tilde{\mathbf{u}}_{t+1}\|^2$.

First, we get that $\|\mathbf{w}_{t+1} - \mathbf{w}_t\|^2 \leq \eta_t^2 \|\sum_{k=1}^p \mathbf{M}_{t,k}(\mathbf{g}_{t,k} + \mathbf{u}_{t,k})\|^2$. According to the definition of $\mathbf{M}_{t,k}$, we have $\forall \gamma > 0$,

$$
\begin{aligned}
\|\mathbf{w}_{t+1} - \mathbf{w}_t\|^2 \leq & \eta_t^2 \|\tilde{\mathbf{g}}_t + \tilde{\mathbf{u}}_t\|^2 \\
= & \eta_t^2 \|\beta(\mathbf{w}_t - \mathbf{w}_{t-1})/\eta_t + \nabla f(\mathbf{w}_t; \mathcal{I}_t) + \tilde{\mathbf{u}}_t\|^2 \\
\leq & (1+\gamma)\beta^2 \|\mathbf{w}_t - \mathbf{w}_{t-1}\|^2 + (1 + \frac{1}{\gamma})(2G^2 + 2U^2)\eta_t^2.
\end{aligned}
$$

By setting $\gamma = \frac{1-\beta}{\beta}$, we obtain $\|\mathbf{w}_{t+1} - \mathbf{w}_t\|^2 \leq \beta\|\mathbf{w}_t - \mathbf{w}_{t-1}\|^2 + \frac{2G^2 + 2U^2}{1-\beta}\eta_t^2$. Then using the proof in Lemma 2, we obtain the first inequality that $\mathbb{E}\|\mathbf{x}_t - \mathbf{w}_t\|^2 \leq \mathcal{O}(\eta_t^2)$.

Next, we get the relation between $\mathbf{x}_{t+1}$ and $\mathbf{x}_t$. According to the definition of $\mathbf{x}_t$, we have

$$
\begin{aligned}
\|\mathbf{x}_{t+1} - \mathbf{w}^*\|^2 &= \|\mathbf{w}_{t+1} + \frac{\beta}{1-\beta}(\mathbf{w}_{t+1} - \mathbf{w}_t) - \frac{\eta_{t+1}}{1-\beta}\tilde{\mathbf{u}}_{t+1} - \mathbf{w}^*\|^2 \\
&= \frac{1}{(1-\beta)^2}\|\mathbf{w}_{t+1} - (\eta_{t+1}\tilde{\mathbf{u}}_{t+1} + \beta\mathbf{w}_t + (1-\beta)\mathbf{w}^*)\|^2.
\end{aligned}
$$

Since $\|\tilde{\mathbf{u}}_t\| \leq U$ uniformly and $\|\mathbf{w}^*\| \leq \frac{W}{2}$, $\eta_t \leq \eta_0 \leq \frac{(1-\beta)W}{2U}$, we get that

$$
\|\eta_{t+1}\tilde{\mathbf{u}}_{t+1} + \beta\mathbf{w}_t + (1-\beta)\mathbf{w}^*\| \leq \eta_{t+1}U + \beta W + (1-\beta)\frac{W}{2} \leq W.
$$

Then we obtain $\eta_{t+1}\tilde{\mathbf{u}}_{t+1} + \beta\mathbf{w}_t + (1-\beta)\mathbf{w}^* \in \Omega$ and

$$
\begin{aligned}
\|\mathbf{x}_{t+1} - \mathbf{w}^*\|^2 \leq & \frac{1}{(1-\beta)^2}\|\mathbf{w}_t - \eta_t \sum_{k=1}^p \mathbf{M}_{t,k}(\mathbf{g}_{t,k} + \mathbf{u}_{t,k}) - (\eta_{t+1}\tilde{\mathbf{u}}_{t+1} + \beta\mathbf{w}_t + (1-\beta)\mathbf{w}^*)\|^2 \\
= & \frac{1}{(1-\beta)^2}\|\mathbf{w}_t - \eta_t(\tilde{\mathbf{g}}_t + \tilde{\mathbf{u}}_t - \tilde{\mathbf{u}}_{t+1}) - (\eta_{t+1}\tilde{\mathbf{u}}_{t+1} + \beta\mathbf{w}_t + (1-\beta)\mathbf{w}^*)\|^2 \\
= & \|\mathbf{x}_t - \mathbf{w}^* - \frac{\eta_t}{1-\beta}\nabla f(\mathbf{w}_t; \mathcal{I}_t) + \frac{\eta_t - \eta_{t+1}}{1-\beta}\tilde{\mathbf{u}}_{t+1}\|^2.
\end{aligned}
$$

Then similar to the analysis in Theorem 1 and Theorem 2, we can prove the convergence for strongly convex and convex cases.

