# OpenReview forum: "Global Momentum Compression for Sparse Communication in Distributed SGD"
_ICLR.cc/2020/Conference — Reject_

### Official Review · AnonReviewer3 · 2019-10-14
**Official Blind Review #3**

**Rating:** 3

**Review:**

Gradient sparsification is an important technique to reduce the communication overhead in distributed training. In this paper, the authors proposed a training method called global momentum compression (GMC) for distributed momentum SGD with sparse gradient. Following existing gradient sparsification techniques such DGC, GMC is also built up on the memory gradient approach; the major distinction between GMC and existing techniques is that GMC keeps track of global gradient to maintain the memory gradient, while the existing technique keeps track of worker-local gradients for memory gradient. The primary contributions in the paper are as the following:

1. The authors propose GMC, a training method for distributed momentum SGD with sparse gradient communication. It uses global gradient (but still achieve sparse communication) to maintain the gradient memory while existing approaches such as DGC use worker-local gradient to do so.

2. The authors prove the convergence rate of GMC for 1. strongly convex and smooth functions 2. convex functions and 3. Non-convex Lipschitz smooth functions. This is the first work on proving the convergence rate of distributed momentum SGD using sparse communication techniques based on memory gradient.

3. Empirically, the authors show that GMC can empirically attain the same model accuracy as conventional distributed momentum SGD with ~100x reduction in communication overhead. It can also match the performance of DGC at the same communication compression rate.

I think in general the ideas and efforts of the authors in proving the convergence rate of distributed momentum SGD with *gradient sparsification* is interesting and important. However, I have the some questions and concerns on validating the claims in the paper. I currently give weak reject but I am happy to raise the score if the authors can clarify or improve in their rebuttal / future drafts. The primary questions and concerns (critical to the rating) are:

1. One important claimed advantage of GMC over existing method is that it uses global gradient for memory gradient, while existing methods such as DGC uses local-work gradient to do so. But I did not find convincing support of this advantage in the paper: Empirically, in the experiment results, I don't think GMC demonstrate better performance than DGC in a statistical meaningful way; instead they are basically demonstrating matching performance. Theoretically, I am not sure if only the global gradient enables the proof of convergence rate while the worker-local gradient cannot. My preliminary feeling is that by bounding the gradient variance, it should also be possible to prove a rate for DGC using worker-local gradient; this is because the difference between the global gradient and the local gradient might be bounded via the gradient variance.

2. In the experiments, the authors focus on momentum SGD for image classification tasks. To better support the versatility and efficacy of GMC, it would be interesting to include some experiments for other domains (e.g. using the STOA transformer style models for NLP tasks). In these models, momentum-like components are also used in the optimizer (e.g. Adam for fairseq for machine translations), it will be interesting to see if the efficacy of GMC also empirically transfer to these settings.

Minor questions (influencing the rating in a secondary way)

1. Regarding the assumptions in the paper, I think assumption 2 need some validation / support to show that it is a proper one. My preliminary feeling is that assumption 2 is intuitive as the sparsification procedures only zero out small values so that the error introduced in the gradient is small and bounded. But it should be more convincing to empirically show the magnitude of u comparing to the magnitude of gradient g in Equ. 8.

2. I notice that the experiments uses conventional momentum SGD for a few epochs as warm up, is there any specific reasoning on using this warmup approach instead of the sparsity level warmup as used in DGC?

3. In the experiments, GMC does not use the factor masking trick while DGC uses. If it is for demonstrating the benefits of global gradient for gradient memory, I think it is more proper to also include the results of DGC without factor masking? In this way, this question can be directly answered in an ablation study way by eliminating the possible contribution of using/not using factor masking.


NITS to improve the paper (not related to the rating):

1. The last contribution bullet forgets to mention that it is about comparing to DGC.

2. In algorithm 1, it is clearer to mention how the mask m is generated (e.g. based on magnitude).

3. In the second paragraph in section 3.2, the vector inner product is not properly written between coefficients and w.

4. Above theorem 1, in the text, the condition for the discussion on the two cases are confusing.

5. In the definition of CR in the first paragraph of section 5, why the summation starts from 5. The text describes as warm up with 5 *epochs* while in the equation it is saying warm up with 4 *steps*.









**Experience Assessment:**

I have read many papers in this area.

**Review Assessment: Checking Correctness Of Derivations And Theory:**

I assessed the sensibility of the derivations and theory.

**Review Assessment: Checking Correctness Of Experiments:**

I assessed the sensibility of the experiments.

**Review Assessment: Thoroughness In Paper Reading:**

I read the paper at least twice and used my best judgement in assessing the paper.

---

### Official Review · AnonReviewer1 · 2019-10-23
**Official Blind Review #1**

**Rating:** 3

**Review:**

This paper proposes a scheme for incorporating compressed (sparsified) gradients with momentum in distributed SGD. The approach differs from others in the literature, comes with theoretical guarantees, and improved performance. The results are correct, and the experiments illustrate that the proposed approach can make a difference (albeit, modest) in the quality of the resulting model.

The main point I find dissatisfying about the theoretical results of the paper are the use of Assumption 2. The memory vector is a parameter of the algorithm. I realize that one can enforce this with a projection, as argued in the paragraph rationalizing this assumption. However, that specific case isn't analyzed and it isn't clear how incorporating that projection would affect the accuracy, since it would essentially be countering the effect of error feedback.

I also find Assumption 3 to be strange. In the convex setting, one can typically show that this follows from Assumption 1 alone under the additional assumption of a suitably small step size. In the non-convex setting it isn't clear what this means, since w^* is not well defined (if there are multiple global minimizers).

Assumption 1 is also strong. Typically one assumes that the stochastic gradients are unbiased, and either that the expected gradient is Lipschitz continuous (in the smooth case), or the expected gradient is bounded (in the non-smooth case). Assuming that the stochastic gradients are uniformly bounded essentially implies that the noise vanishes when the gradient gets large.

Can you provide examples of functions/problems satisfying these assumptions? Even an example as simple as the case where $F$ is a finite sum of quadratic functions, and one randomly samples one of the terms in the finite sum to compute the gradient (i.e., using SGD to solve a large linear least squares problem) doesn't appear to satisfy Assumption 1.

Overall the results are potentially interesting. I would have given a higher rating if the assumptions didn't appear to be so strong, and if the experimental results demonstrated a more substantial difference with DGC.




**Experience Assessment:**

I have read many papers in this area.

**Review Assessment: Checking Correctness Of Derivations And Theory:**

I assessed the sensibility of the derivations and theory.

**Review Assessment: Checking Correctness Of Experiments:**

I carefully checked the experiments.

**Review Assessment: Thoroughness In Paper Reading:**

I read the paper at least twice and used my best judgement in assessing the paper.

---

### Official Review · AnonReviewer2 · 2019-10-23
**Official Blind Review #2**

**Rating:** 3

**Review:**

The author propose a method called global momentum compression for sparse communication setting. The contributtions can be summarized into 3 parts: switching DGC setting from local momentum to global momentumm, theortical proof of the convergence, empirical results showing performance.

However there have several issues:
	1. No significant contribution. Although they theoretically prove a new version of DGC, it's just a minor modification and no significant performance improvement as shown from their empirical results.
	2. In the experiment session, as shown in the results,  their method seems more stable during training but there achieves minor improvement in terms of the test accurarcy. Second, they only compare with DGC and it's counterpart baseline. It's better to include more related algorithms for comparison (like quantization methond: QSGD or signSGD).
	3. Comparing with DGC, there is no improvement in saving communication as shown in their results. Since GMC only changes DGC setting from local momentum to global momentum, no modification is involved in the compression part of DGC.

Overall, I appreciate the authors for their theoretical contribution for DGC and well written paper. However, it would be great to show a better improvement and include more related methods for comparison.

**Experience Assessment:**

I have published one or two papers in this area.

**Review Assessment: Checking Correctness Of Derivations And Theory:**

I did not assess the derivations or theory.

**Review Assessment: Checking Correctness Of Experiments:**

I assessed the sensibility of the experiments.

**Review Assessment: Thoroughness In Paper Reading:**

I made a quick assessment of this paper.

---

### Decision · Program_Chairs · 2019-12-19

**Decision:**

Reject

**Comment:**

The author propose a method called global momentum compression for sparse communication setting, and provided some theoretical results on the convergence rate. The convergence result is interesting, but the underlying assumptions used in the analysis appear very strong. Moreover, the proposed algorithm has limited novelty as it is only a minor modification. Another main concern is that the proposed algorithm shows little performance improvement in the experiments. Moreover, more related algorithms should be included in the experimental comparison.